# Metabolic diversity and niche structure in sponges from the Miskito Cays, Honduras

Christopher J. Freeman[1], Cole G. Easson[2] and David M. Baker[3]

[1] Smithsonian Marine Station, Fort Pierce, FL, USA
[2] Department of Biology, University of Alabama at Birmingham, Birmingham, AL, USA
[3] School of Biological Sciences and Swire Institute of Marine Science, University of Hong Kong, Hong Kong, PRC

Corresponding author
Christopher J. Freeman,
freemanc@si.edu

## ABSTRACT

Hosting symbionts provides many eukaryotes with access to the products of microbial metabolism that are crucial for host performance. On tropical coral reefs, many (High Microbial Abundance [HMA]) but not all (Low Microbial Abundance [LMA]) marine sponges host abundant symbiont communities. Although recent research has revealed substantial variation in these sponge-microbe associations (termed holobionts), little is known about the ecological implications of this diversity. We investigated the expansion of diverse sponge species across isotopic niche space by calculating niche size (as standard ellipse area [$SEA_c$]) and assessing the relative placement of common sponge species in bivariate ($\delta^{13}C$ and $\delta^{15}N$) plots. Sponges for this study were collected from the relatively isolated reefs within the Miskito Cays of Honduras. These reefs support diverse communities of HMA and LMA species that together span a gradient of photosymbiont abundance, as revealed by chlorophyll *a* analysis. HMA sponges occupied unique niche space compared to LMA species, but the placement of some HMA sponges was driven by photosymbiont abundance. In addition, photosymbiont abundance explained a significant portion of the variation in isotope values, suggesting that access to autotrophic metabolism provided by photosymbionts is an important predictor in the location of species within isotopic space. Host identity accounted for over 70% of the variation in isotope values within the Miskito Cays and there was substantial variation in the placement of individual species within isotopic niche space, suggesting that holobiont metabolic diversity may allow taxonomically diverse sponge species to utilize unique sources of nutrients within a reef system. This study provides initial evidence that microbial symbionts allow sponges to expand into novel physiochemical niche space. This expansion may reduce competitive interactions within coral reefs and promote diversification of these communities.

## INTRODUCTION

By expanding the metabolic repertoires of foundational species, symbiotic interactions represent important processes underlying some of the most biodiverse ecosystems

worldwide (*Moya et al., 2008*; *Vrijenhoek, 2010*). This is exemplified on oligotrophic coral reefs, where the autotrophic metabolism of microbial symbionts supplements heterotrophic feeding of reef-building corals (*Muscatine & Cernichiari, 1969*; *Boucher, James & Keeler, 1982*; *Muscatine, Porter & Kaplan, 1989*; *Baker et al., 2013*). Interestingly, although it is well established that severe nutrient limitation has favored the evolution of these and other host-symbiont units (termed holobionts; *Rosenberg et al., 2007*; *Zilber-Rosenberg & Rosenberg, 2008*), it is becoming increasingly apparent that evolution has also led to diversification in these interactions across functional groups and host species (*Thacker & Freeman, 2012*; DM Baker, 2015, unpublished data).

For instance, while some sponge hosts support abundant and diverse symbiont communities (termed High Microbial Abundance [HMA]), other sympatric sponge species only host sparse symbiont communities (termed Low Microbial Abundance [LMA]) (*Webster et al., 2010*; *Schmitt et al., 2012*; *Thacker & Freeman, 2012*; *Gloeckner et al., 2014*). By hosting autotrophic symbionts, many of these HMA species are able to assimilate both inorganic and organic sources of C and N, whereas LMA sponges may be restricted to feeding heterotrophically on specific portions of the particulate organic matter (POM) pool in the water column (*Thurber, 2007*; *Taylor et al., 2007*; *Maldonado, Ribes & van Duyl, 2012*; *Thacker & Freeman, 2012*). The placement of sponges into one of these putative guilds is widespread, but symbiont metabolism is also highly variable across different HMA species, making such broad generalizations difficult. For instance, *Freeman & Thacker (2011)* demonstrated that 3 HMA species each had a unique interaction with their respective symbiont community and this was ultimately ascribed to the presence or absence of productive photosymbiont taxa across different sponge species (*Freeman et al., 2013*). In addition, with increasing evidence supporting the assimilation of dissolved organic matter (DOM) (*Maldonado, Ribes & van Duyl, 2012*; *de Goeij et al., 2013*), it is apparent that there is substantial metabolic diversity across sympatric sponge species.

This diversity may allow for the expansion of sponge species across available niche space within crowded tropical reef environments, potentially relaxing competition and contributing to the diversification of these communities (*Knowlton & Jackson, 1994*; *Beinart et al., 2012*; *Joy, 2013*). Unfortunately, our understanding of the relative positions of diverse sponge species within the available niche space of a reef remains limited. The relative placement of an organism within a food web has long been assessed using the stable isotopes of C and N ($\delta^{13}$C and $\delta^{15}$N) (*Fry, 2006*). Because an organism's position in bivariate ($\delta^{13}$C and $\delta^{15}$N) "isotopic space" is directly related to the sources of C and N assimilated, biochemical processing of these sources, and the habitat where an organism is found, the relative placement of a consumer in this space has been equated with the "niche" space it fills within a system (*Layman et al., 2007a*; *Newsome et al., 2007*; *Layman et al., 2012*). Qualitative assessments of an organism's niche are possible with $\delta^{13}$C and $\delta^{15}$N biplots, and these analyses have been widely used to investigate sponge metabolism (*Weisz, 2006*; *Thurber, 2007*; *Weisz et al., 2007*; *Mohamed et al., 2008*; *Freeman & Thacker, 2011*; *Van Duyl et al., 2011*; *Fiore, Baker & Lesser, 2013*). Quantitative assessments of niche space have recently been described and used to study trophic diversity and investigate changes

in the size and structure of a community or population (*Layman et al., 2007a*; *Layman et al., 2007b*; *Layman et al., 2012*). Recent analytical methods also provide a means to evaluate the relative placement of a consumer or guild within isotopic space (*Turner, Collyer & Krabbenhoft, 2010*).

We adapted these methods to calculate the niche size and relative placement of HMA and LMA groups and individual sponge species within isotopic niche space (*Turner, Collyer & Krabbenhoft, 2010*; *Jackson et al., 2011*). Sponges for this study were collected from 14 sites within the Miskito Cays of Honduras during a May 2013 expedition in this archipelago (*Chollett, Stoyle & Box, 2014*). As part of this expedition, sponge communities were surveyed and sampled to gain an understanding of the diversity of conspicuous, common sponge species and to assess the status of sponge-microbe symbioses on these largely undescribed and isolated Caribbean reefs. These reefs are distant from chronic anthropogenic stressors (*Hay, 1984*; *Chollett, Stoyle & Box, 2014*) and thus may provide a baseline understanding of how diverse sponges fill isotopic niche space in oligotrophic systems that can be compared to more eutrophic systems throughout the greater Caribbean. We test the following null hypotheses: (1) niche size is similar between HMA and LMA groups and across individual sponge species, and (2) HMA and LMA groups and individual sponge species occupy similar niche space within a reef system.

## MATERIALS & METHODS

### Study site and collection

The Miskito Cays are an archipelago of 49 small (average 0.59 hectares) islands representing about 750 km$^2$ of benthic habitat located approximately 300 km east of the Bay Islands and 70 km off the northeast coast of Honduras (*Chollett, Stoyle & Box, 2014*). Sponge diversity was assessed qualitatively at 14 sites within this region (Table 1; see map in *Chollett, Stoyle & Box, 2014*) by recording the presence or absence of apparent, non-cryptic sponge species. When possible, at least three replicate samples of each common species were collected at a site for $\delta^{13}$C and $\delta^{15}$N values and chlorophyll *a* (chl *a*) analyses, along with a carboy of water collected at depth for the measurement of $\delta^{13}$C and $\delta^{15}$N values of particulate organic matter (POM), a potential source of C and N for sponges feeding heterotrophically (*Freeman & Thacker, 2011*). Because filtration of water for POM was restricted to the use of a hand pump, collections for POM were limited to a single carboy at each site.

After each dive, sponge samples were catalogued and, because sponges could not be frozen for the duration of the expedition, samples were dried for 36 h at 60 °C using the Nesco FD-75A 700 W food dehydrator. Water samples were filtered through a 0.70 μm GF filter under low pressure generated via a hand pump to obtain POM as in *Freeman & Thacker (2011)*; filters were wrapped in aluminum foil and dried for 24 h in the dehydrator as above. These collections were organized and authorized through an agreement between the Centre for Marine Studies, a Honduran non-governmental organization, and the national government; the Smithsonian Institution provided technical support. These sponge species are not listed within the CITES appendices and are not protected within Honduras.

**Table 1** Names and GPS coordinates of 14 sites visited within the Miskito Cays of Honduras.

| Site | Latitude-N | Longitude-W |
| --- | --- | --- |
| Vivorillos Site #1 | 15.837 | −83.291 |
| Vivorillos Site #2 | 15.863 | −83.306 |
| Becerros Site #1 | 15.913 | −83.255 |
| Becerros Site #2 | 15.951 | −83.272 |
| Caratasca Site #1 | 16.024 | −83.316 |
| Caratasca Site #2 | 16.030 | −83.319 |
| Cajones Site #1 | 16.033 | −83.094 |
| Cajones Site #2 | 16.057 | −83.100 |
| Cajones Site #3 | 16.085 | −83.143 |
| Cajones Site #4 | 16.093 | −83.173 |
| Media Luna Site #1 | 15.261 | −82.631 |
| Media Luna Site #2 | 15.186 | −82.618 |
| Media Luna Site #3 | 15.139 | −82.582 |
| Media Luna Site #4 | 15.122 | −82.587 |

## $\delta^{13}$C and $\delta^{15}$N and Chlorophyll *a* analyses

At the Smithsonian Marine Station in Fort Pierce, Florida, USA, sponge samples were placed in a 60 °C oven to remove any residual moisture and ground to a fine powder using a mortar and pestle. Homogenized sponge tissue was acidified to remove carbonate and weighed into tared silver capsules for $\delta^{13}$C and $\delta^{15}$N as in *Freeman & Thacker (2011)*. Sponge and POM samples were analyzed in the Stable Isotope Ratio Mass Spectrometry laboratory (SIRMS) at the University of Hong Kong via combustion in a Eurovector EA3028 coupled to a Perspective IRMS (Nu Instruments). Analytical precision was determined by repeated analysis of an internal acetanilide standard ('acet 6'; 70% C). Mean ($\pm$SE) precision during analysis was 0.2 $\pm$ 0.04 and 0.1 $\pm$ 0.01 for $\delta^{15}$N and $\delta^{13}$C, respectively. To assess photosymbiont abundance (chl *a*), analyses were carried out on dried tissue as in *Freeman & Thacker (2011)* and *Freeman et al. (2013)*.

## Data analysis

Isotopic niche area of LMA and HMA groups and individual sponge species was estimated by calculation of the standard ellipse area ($SEA_c$) using a Bayesian approach based on multivariate ellipse-based metrics (SIBER [Stable Isotope Bayesian Ellipses in R]; *Jackson et al., 2011*). The $SEA_c$ contains approximately 40% of the data within a set of bivariate data and thus represents the core niche area for a population or community (*Jackson et al., 2011*; *Layman et al., 2012*). Unlike previous estimates of niche width (by measuring the area of a convex hull enclosing all data points [Total Area TA; *Layman et al., 2007a*]), $SEA_c$ calculations are less susceptible to outlying data points. In addition, estimation of these ellipses by Bayesian inference allows for robust comparison across sets of data with different sample sizes. We adapted these methods to first investigate differences in the size and overlap of the $SEA_c$ of LMA and HMA sponges at sites within the Miskito

Cays. Because our goals using $SEA_c$ were to (1) visualize the variation within each of these well-established guilds of sponges and (2) quantify and compare the core niche areas for each of these groups, $SEA_c$ was calculated for each group from $\delta^{13}C$ and $\delta^{15}N$ values of individual sponge samples (*Turner & Edwards, 2012*). Likewise, to compare the niche area of individual species, the $SEA_c$ of each species was calculated by the dispersion of $\delta^{13}C$ and $\delta^{15}N$ values of each species within isotopic space (*Jackson et al., 2011*). For all analyses, residuals from General Linear Models were tested for normality and equal variance using Kolmogorov–Smirnov and Levene's tests, respectively.

In addition to the calculation of $SEA_c$, we compared the placement of HMA and LMA groups and individual species in isotopic niche space using methods outlined by *Turner, Collyer & Krabbenhoft (2010)*. These methods calculate the Euclidean distance between the centroids (bivariate mean) of guilds or individual species within isotopic space and use a residual permutation procedure (RPP) and Hotelling $T^2$ test to evaluate whether this distance is significant (different from zero), thus placing these groups or individual species in unique isotopic space (*Turner, Collyer & Krabbenhoft, 2010*). Finally, to quantify the relative effects of microbial abundance, chlorophyll *a* concentration, and host species identity on the placement of samples with the isotopic space of the Miskito Cays, we calculated isotopic dissimilarity among samples. This was accomplished by creating a distance matrix from our $\delta^{13}C$ and $\delta^{15}N$ data and analyzing sample distances using the R function adonis in the package vegan (*Oksanen et al., 2014*).

## RESULTS

Sponges were sampled at 14 sites spanning over 100 km of the Miskito Cays (Table 1; *Chollett, Stoyle & Box, 2014*). We catalogued 25 species of sponges, with substantial variation in species diversity (Table 2) across sites. Because collections were limited to single dives at each site and not all species were abundant at each site, replicate individuals of a species observed were not always collected. For instance, although site ML4 was extremely diverse, we were only able to collect replicate individuals of five species from this site. Collections at two of the most diverse sites, Media Luna #s 2 and 3 (ML2 and ML3), were the most complete (Table 2), so within site comparisons are focused on these sites. Particulate organic matter (POM) from 6 sites within Miskito Cays had a mean ($\pm$SE) value of 3.7‰ $\pm$ 0.6 and $-22.7$‰ $\pm$ 0.5 for $\delta^{15}N$ and $\delta^{13}C$, respectively, with a range in $\delta^{15}N$ values of 0.9‰ (Site ML4) to 5.5‰ (Vivorillos Site 1) and $\delta^{13}C$ values of $-24.1$‰ (Site ML2) to $-21.2$‰ (Site ML4). We were unable to obtain accurate C and N values of POM at some sites due to low signal strength resulting from low organic biomass on filters.

Photosymbiont abundance (as measured by chlorophyll *a* [chl *a*]) was highly variable across the 19 most common sponge species collected within this region. Of these species, 10 have been previously categorized as HMA and 8 as LMA (*Weisz, 2006*; *Weisz et al., 2007*; *Thacker & Freeman, 2012*; *Maldonado, Ribes & van Duyl, 2012*; *Gloeckner et al., 2014*). Although *Desmapsamma anchorata* has yet to be placed within one of these groups, because this species is characterized by low chl *a* concentration and lacks defined symbionts (Fig. 1; *Erwin & Thacker, 2007*), it has been included within the LMA group.

**Table 2 Species of conspicuous sponges observed (X) at 14 sites within the Miskito Cays of Honduras.**

| Sponge species | Vivorillos Site #1 | Vivorillos Site #2 | Becerros Site #1 | Becerros Site #2 | Caratasca Site #1 | Caratasca Site #2 | Cajones Site #1 | Cajones Site #2 | Cajones Site #3 | Cajones Site #4 | Media Luna Site #1 | Media Luna Site #2 | Media Luna Site #3 | Media Luna Site #4 |
|---|---|---|---|---|---|---|---|---|---|---|---|---|---|---|
| *Agelas conifera* (Schmidt, 1870) | | | | | | | | | | | | X | X | X |
| *Agelas wiedenmayeri* Alcolado, 1984 | X | | X | | | | | | | | | | | |
| *Aiolochroia crassa* (Hyatt, 1875) | | | | | | | | | | | | X | X | X |
| *Siphonodictyon coralliphagum* Rützler, 1971 | | | | | | | | | | | | X | | |
| *Amphimedon compressa* Duchassaing & Michelotti, 1864 | | | X | | X | | | | | | X | X | X | X |
| *Aplysina cauliformis* (Carter, 1882) | | X | | X | X | X | | | | | X | X | X | X |
| *Aplysina fulva* (Pallas, 1766) | | X | | | X | X | | | | | X | X | X | X |
| *Aplysina lacunosa* (Lamarck, 1814) | | | | | | | | | | | | X | X | X |
| *Callyspongia (Callyspongia) fallax* Duchassaing & Michelotti, 1864 | | | | X | X | | | | | | | X | X | X |
| *Callyspongia (Cladochalina) plicifera* (Lamarck, 1814) | | | | | X | | | | | | | X | X | X |
| *Callyspongia (Cladochalina) vaginalis* (Lamarck, 1814) | | X | | | X | | | | | | X | X | X | X |
| *Chondrilla caribensis* Rützler, Duran, & Piantoni, 2007 | | | | | | | | | | | | X | X | X |
| *Cliona delitrix* Pang, 1973 | | | | | | | | | | | | X | X | |
| *Desmapsamma anchorata* (Carter, 1882) | | | | | | | | | | | | | X | X |

Table 2 (*continued*)

| Sponge species | Vivorillos Site #1 | Vivorillos Site #2 | Becerros Site #1 | Becerros Site #2 | Caratasca Site #1 | Caratasca Site #2 | Cajones Site #1 | Cajones Site #2 | Cajones Site #3 | Cajones Site #4 | Media Luna Site #1 | Media Luna Site #2 | Media Luna Site #3 | Media Luna Site #4 |
|---|---|---|---|---|---|---|---|---|---|---|---|---|---|---|
| *Ectyoplasia ferox* (Duchassaing & Michelotti, 1864) | X | | | | | | | | | | | | X | |
| *Iotrochota birotulata* (Higgin, 1877) | | X | | X | X | X | | | | | | X | X | X |
| *Ircinia campana* (Lamarck, 1814) | | | | X | X | | | | | | | X | X | X |
| *Ircinia felix* (Duchassaing & Michelotti, 1864) | X | | | | | | | | | | | X | X | X |
| *Ircinia strobilina* (Lamarck, 1816) | | | | | | | | | | | | X | | X |
| *Monanchora arbuscula* (Duchassaing & Michelotti, 1864) | X | | | | | | | | | | | X | X | X |
| *Mycale (Mycale) laevis* (Carter, 1882) | | | | | X | | | | | | | X | X | X |
| *Neopetrosia rosariensis* (Zea & Rützler, 1983) | X | | | X | | | | | | | | | | |
| *Neopetrosia subtriangularis* (Duchassaing, 1850) | | | | | | | | | | | | | | X |
| *Niphates erecta* (Duchassaing & Michelotti, 1864) | | X | X | | | | | | | | | X | X | X |
| *Verongula rigida* (Esper, 1794) | X | X | X | | X | X | | | | | X | X | X | X |

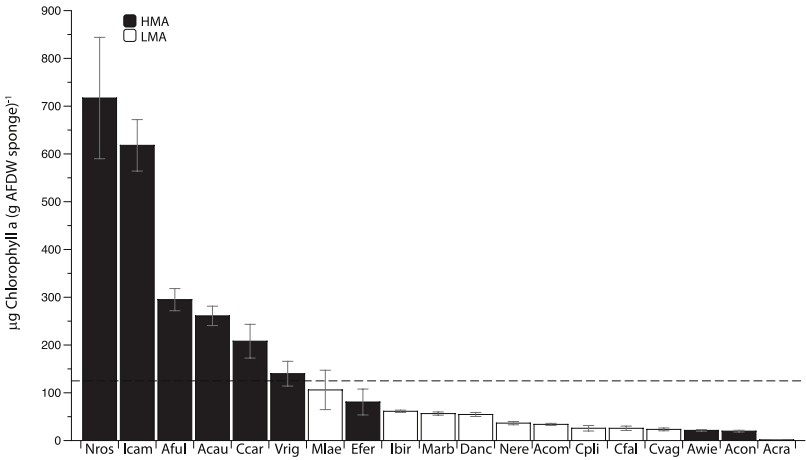

**Figure 1 Mean Chlorophyll *a* (Chl *a*) concentration (±SE) of 19 common sponge species collected within the Miskito Cays of Honduras.** Sponges are categorized based on their overall microbial abundance into High Microbial Abundance (HMA) and Low Microbial Abundance (LMA) groups (*Weisz, 2006*; *Erwin & Thacker, 2007*; *Maldonado, Ribes & van Duyl, 2012*; *Gloeckner et al., 2014*). The horizontal black line represents the cutoff above which sponges are considered to have high chl *a* (>125 μg of chl *a* [g of sponge tissue]$^{-1}$; *Erwin & Thacker, 2007*). Abbreviations represent the first letter of the genus name, followed by the first three letters of the specific epithet.

Six of the 19 species (all of which are HMA species) are considered to have high chl *a* concentrations (chl *a* >125 μg of chl *a*; as in *Erwin & Thacker, 2007*), while all of the LMA species were well below this cutoff. Four of the 10 HMA species had low concentrations of chl *a* (Fig. 1; *Gloeckner et al., 2014*).

Although SIBER analyses showed a 31% overlap of HMA and LMA species within the isotopic niche space of the Miskito Cays, HMA sponges displayed a broader niche width (greater SEA$_c$) than LMA sponges ($P < 0.0001$) (Fig. 2A). We have included the TA metric (Total Area of convex hulls) in the figures as a reference, but the remaining results and discussion will focus solely on SEA$_c$ as a measure of niche area, which is less sensitive to variation in sample size than TA based on convex hulls (Fig. 2A; *Layman et al., 2007a*; *Jackson et al., 2011*). Each of these two groups fills unique isotopic niche space within the Miskito Cays (distance between centroids = 1.45; Hotelling's $T$ test: $T^2 = 119.21$, $F = 58.85$, $P < 0.0001$) (Fig. 2A). When HMA sponges are further delineated into species with high (HMA-H) and low (HMA-L) chl *a* values, the SEA$_c$ of HMA-H species is significantly larger than that of both HMA-L and LMA groups ($P < 0.01$) (Fig. 2B). In addition, HMA-H sponges occupied a unique location in isotopic niche space compared to both HMA-L (57% SEA$_c$ overlap; distance between centroids = 1.50; Hotelling's $T$ test: $T^2 = 43.68$, $F = 21.28$, $P < 0.001$) and LMA sponges (<0.1% SEA$_c$ overlap; distance between centroids = 1.71; Hotelling's $T$ test: $T^2 = 177.21$, $F = 87.38$, $P < 0.001$). Although the SEA$_c$ of the HMA-L species was significantly larger than that of the LMA group ($P = 0.029$), LMA and HMA-L species occupied similar isotopic niche space (66.7% SEA$_c$ overlap; distance between centroids = 0.25; Hotelling's $T$ test: $T^2 = 1.44$, $F = 0.70$, $P = 0.492$) (Fig. 2B). The placement of individual samples within the isotopic space of the Miskito Cays was significantly impacted by microbial abundance (adonis effect

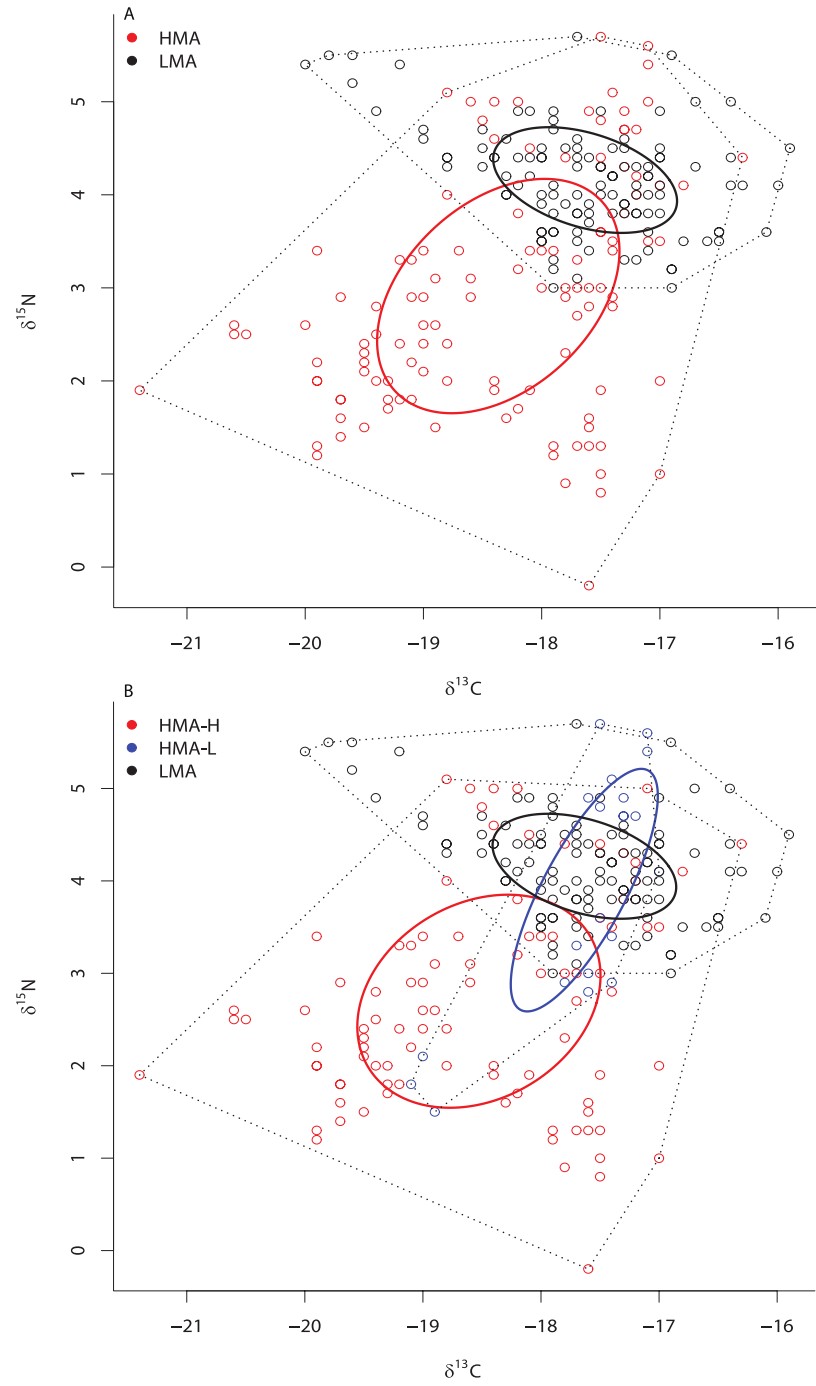

**Figure 2 Bivariate $\delta^{13}$C and $\delta^{15}$N plot depicting the placement of HMA and LMA (A) and HMA sponges with high and low chl *a* concentrations (HMA-H and HMA-L, respectively) and LMA groups (B) within the isotopic niche space of the Miskito Cays.** Standard ellipse areas (SEA$_c$) depicted by solid lines provide estimates of the niche area of each of these groups using Bayesian inference as in *Jackson et al. (2011)*. Convex hulls depicting total niche width of each group are shown using dashed lines for reference *Layman et al. (2007a)*.

of HMA/LMA: $df = 1$, $F = 62.12$, $R^2 = 0.21$, $P < 0.01$), but was better explained by modified microbial abundance (adonis effect of HMA-H, HMA-L, and LMA: $df = 2$, $F = 44.59$, $R^2 = 0.27$, $P < 0.01$). An equal amount of variance in sample placement within isotopic space was explained by chlorophyll $a$ concentration alone (adonis effect of chl $a$ concentration (high or low): $df = 1$, $F = 88.57$, $R^2 = 0.27$, $P < 0.01$). In addition, when chlorophyll $a$ concentration and modified microbial abundance were tested together in the same equation using adonis, it was apparent that these factors explained the same portion of the variance, implying that chlorophyll $a$ concentration was the main explanatory factor for determining sponge placement within isotopic space.

At ML2, the $SEA_c$ of HMA species was also significantly larger than that of LMA species ($P < 0.01$), there was no overlap in the $SEA_c$ of these two groups, and each of these groups occupied unique niche space (distance between centroids = 1.84; Hotelling's $T$ test: $T^2 = 85.95$, $F = 40.50$, $P < 0.0001$) (Fig. 3A). Likewise, at site ML3, the $SEA_c$ of HMA species was significantly larger than that of LMA species (0% of the simulated LMA $SEA_c$ were greater than those of HMA holobionts) (Fig. 3B) and each of these two groups occupied unique isotopic niche space (distance between centroids = 1.39; Hotelling's $T$ test: $T^2 = 24.68$, $F = 11.55$, $P < 0.0001$). At site ML3, there was substantial (31.7%) overlap in the $SEA_c$ generated by LMA and HMA holobionts. Because collections at ML3 included members of both HMA-H and HMA-L species, we repeated this analysis with the three groups (HMA-L, HMA-H, and LMA). Although the $SEA_c$ of HMA-H and HMA-L groups were similar in size ($P = 0.671$), the $SEA_c$ of both HMA groups were greater than that of LMA sponges ($P < 0.007$). While there was no overlap between HMA-H and either HMA-L or LMA sponges, these latter two groups overlapped by almost 40% (Fig. 3C). Each of these 3 groups occupied unique isotopic niche space within ML3 (HMA-H and HMA-L: distance between centroids = 1.89; Hotelling's $T$ test: $T^2 = 23.85$, $F = 9.69$, $P = 0.001$; HMA-H and LMA: distance between centroids = 2.55; Hotelling's $T$ test: $T^2 = 58.08$, $F = 25.81$, $P < 0.0001$; HMA-L and LMA: distance between centroids = 0.71; Hotelling's $T$ test: $T^2 = 7.19$, $F = 3.15$, $P = 0.046$).

Separate SIBER analysis on each of the 19 sponge species from across all sites within the Miskito Cays increased our resolution of the dispersion of different holobionts within isotopic space (Fig. 4). Although pairwise differences in the size of $SEA_c$ were widespread, we limit our discussion of these data, as some but not all species were collected from disparate sites. Within the isotopic niche space of the Miskito Cays, 4 of the 6 HMA-H species (*Aplysina* spp., *I. campana*, and *C. caribensis*) were isolated from LMA sponges, while the other 2 HMA-H species (*N. rosariensis* and *V. rigida*) overlapped with one or more LMA species. Likewise, the HMA-L *Agelas* spp. overlapped with at least one LMA sponge, while the HMA-L species *E. ferox* overlapped with *A. cauliformis* and was distant in space from LMA species. The ellipses representing the LMA species were concentrated in one region of isotopic niche space, with substantial overlap, whereas the $SEA_c$ of HMA species were more widespread (Fig. 4). Host species identity accounted for nearly 75% of the variation in isotope values across all individuals (adonis effect of species: $df = 18$, $F = 34.38$, $R^2 = 0.74$, $P < 0.01$).

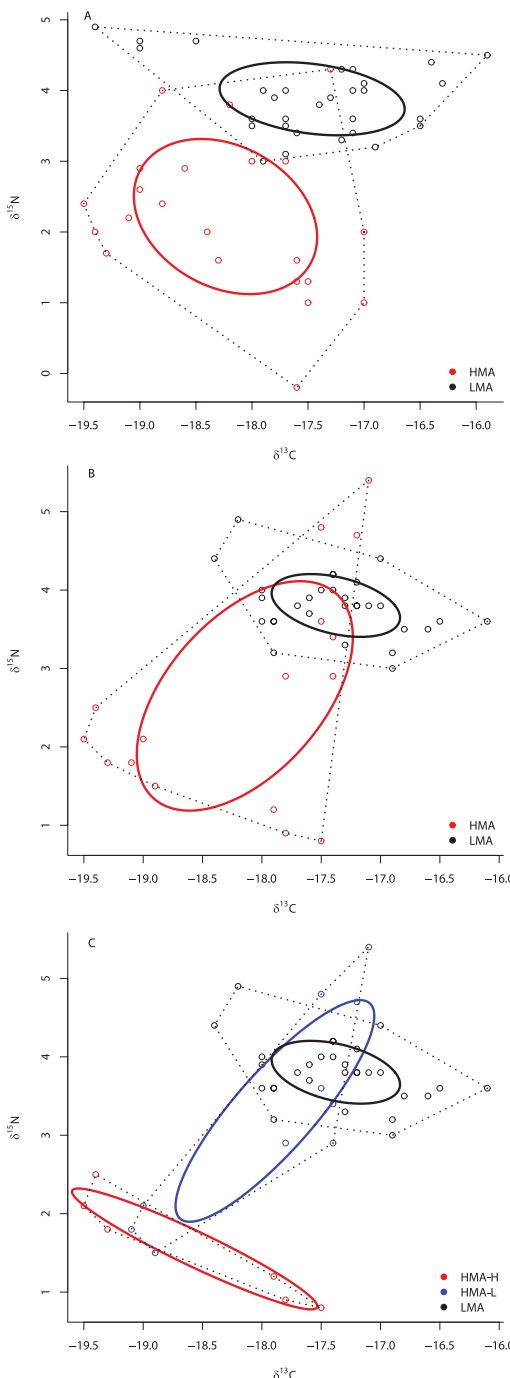

**Figure 3** **Bivariate $\delta^{13}$C and $\delta^{15}$N plot depicting the placement of HMA and LMA sponges within the isotopic niche space of Media Luna Site #2 (A) and Media Luna Site #3 (B).** In addition, (C) depicts the placement of HMA sponges with high and low chl *a* concentrations (HMA-H and HMA-L, respectively) and LMA groups within the isotopic niche space of Media Luna Site #3 (C). As in Fig. 2, standard ellipse areas (SEA$_c$) are depicted by solid lines and convex hulls are depicted using dashed lines for reference. Note the difference in the scale of the $\delta^{15}$N axis between (A) and (B) and (C).

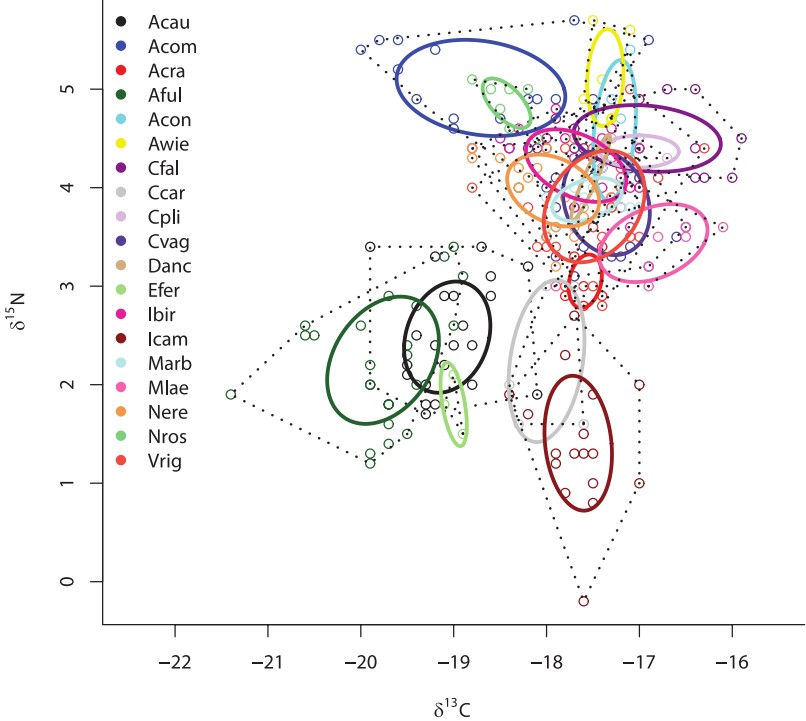

**Figure 4 Bivariate $\delta^{13}$C and $\delta^{15}$N plot depicting the placement of 19 sponge species within the isotopic niche space of the Miskito Cays.** As in Fig. 2, solid lines depict standard ellipse areas (SEA$_c$) and dashed lines depict convex hulls for reference. Species abbreviations are the same as in Fig. 1.

Although there was no significant difference in the size of the SEA$_c$ across species within sites ML2 and ML3 due to limited statistical power in SIBER as a result of low sample size, there was substantial variation in the placement of each species within isotopic niche space within these sites (Figs. 5A and 5B). Within ML2, some species were present in unique niche space (*I. campana*, *C. caribensis*, *C. fallax*, and *A. compressa*), while other species shared niche space with at least one other species (the HMA *Aplysina spp*; the HMA species *V. rigida* and the LMA species *M. arbuscula* and *N. erecta*; and the LMA species *C. vaginalis* and *I. birotulata* and *M. laevis*; Fig. 5A). Similar results were observed in the 11 species collected from ML3 (Fig. 5B), where the HMA species *A. cauliformis*, *E. ferox* and *I. campana* each occupied a unique location within isotopic space at this site, while other species shared niche space with at least one other species.

## DISCUSSION

This study supports the contention that biogeochemical cycling of C and N is highly variable across sponge species and provides initial evidence that this variation is driven more by host species identity than by overall symbiont abundance. These reefs within the Miskito Cays of Honduras support diverse sponge communities that are taxonomically similar to those of other Caribbean reefs (*Díaz, 2005*; *Erwin & Thacker, 2007*; *Chollett, Stoyle & Box, 2014*) and include species hosting both abundant and sparse microbial taxa (HMA and LMA, respectively; *Thacker & Freeman, 2012*). In addition, species from this

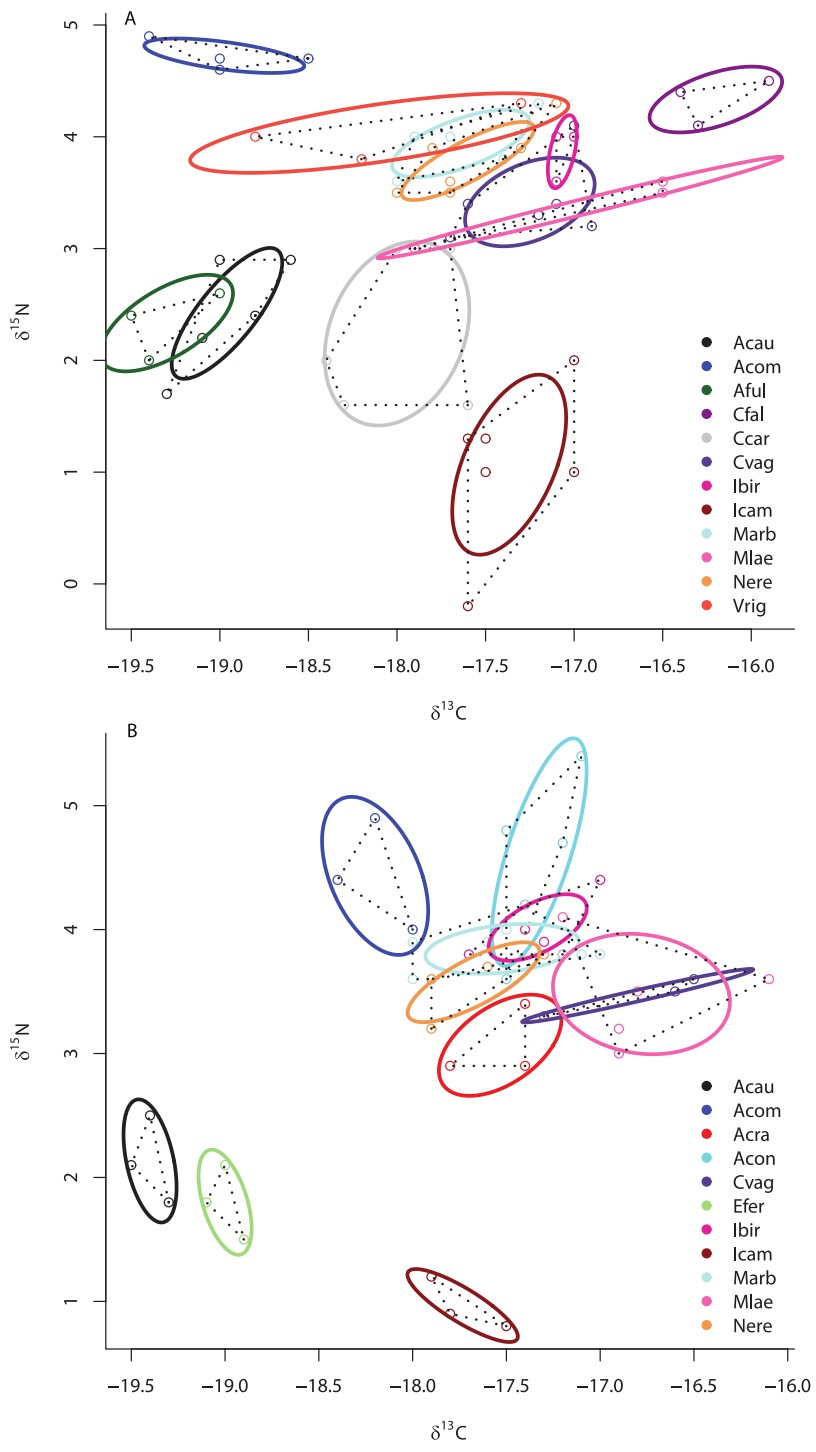

**Figure 5** Bivariate $\delta^{13}$C and $\delta^{15}$N plot depicting the placement of 12 species within the isotopic niche space of Media Luna Site #2 (A) and 11 species within the isotopic niche space of Media Luna Site #3 (B). As in Fig. 2, standard ellipse areas (SEA$_c$) are depicted by solid lines and convex hulls are depicted using dashed lines for reference. Species abbreviations are the same as in Fig. 1. Note the difference in the scale of the $d^{15}$N axis between (A) and (B).

region spanned a striking gradient of photosymbiont abundance (*Erwin & Thacker, 2007*) with some, but not all HMA sponges support abundant photosymbiont communities.

Symbionts within HMA species appear to increase holobiont metabolic diversity, leading to broader niche area (SEA$_c$) than that observed in LMA species (*Boucher, James & Keeler, 1982*; *Vrijenhoek, 2010*). In addition, the separation of HMA and LMA groups suggests that biogeochemical cycling of C and N is distinct between HMA and LMA species and that, by hosting abundant symbiont communities, HMA species expand into novel isotopic niche space (*Weisz et al., 2007*). Although microbial abundance accounted for a significant portion of the variance in isotopic values across the Miskito Cays, this variance was better explained by chlorophyll *a* concentrations, implying that photosymbiont metabolism is an important predictor in the placement of species in isotopic space (*Wilkinson, 1983*; *Stanley, 2006*; *Venn, Loram & Douglas, 2008*). While species lacking photosymbionts may be largely restricted to heterotrophic feeding to meet their energy demands, abundant microbial communities within HMA-L sponges may assimilate diverse sources of DOM (*Maldonado, Ribes & van Duyl, 2012*) or may utilize chemoautotrophic pathways to fix nutrients (*Taylor et al., 2007*). With increasing evidence that sponge cells can directly assimilate DOM and that many LMA species have moderate OTU richness, supplemental nutrition via DOM may also occur in some LMA species (*de Goeij et al., 2013*; *Easson & Thacker, 2014*).

Species accounts for a significant (~74%) portion of the variance in isotope values within the Miskito Cays, suggesting that biogeochemical C and N cycling is highly variable across host species. Indeed, the unique placement of individual species within isotopic niche space may reflect the assimilation of unique sources of C and N or species-specific biological and chemical processes occurring during the transformation and cycling of these nutrients within the holobiont (*Weisz, 2006*; *Newsome et al., 2007*; *Taylor et al., 2007*; *Southwell, Popp & Martens, 2008*; *Fiore et al., 2010*; *Layman et al., 2012*; *Fiore, Baker & Lesser, 2013*). For instance, some holobionts may fix and further transform inorganic sources of C and N (*Mohamed et al., 2008*; *Maldonado, Ribes & van Duyl, 2012*; *Freeman et al., 2013*), while other holobionts could be utilizing a combination of inorganic and organic sources or relying solely on local sources of dissolved or particulate organic matter (*Van Duyl et al., 2011*; *Maldonado, Ribes & van Duyl, 2012*; *Thacker & Freeman, 2012*). In addition, disparity in the placement of species could be reflective of specialization on particular size fractions of the pelagic microbial loop (*Thurber, 2007*) or variation in the proportional use of diverse DOM sources (coral mucus, phytoplankton, and benthic algae [including crustose coralline algae-CCAs] *Van Duyl et al., 2011*). Coupling these analyses with comprehensive sampling of potential sources of nutrients, feeding experiments with labeled organic and inorganic sources of C and N, and fatty acid analysis may help to better elucidate resource use by diverse holobionts in reef systems (*Thurber, 2007*; *Van Duyl et al., 2011*; *Maldonado, Ribes & van Duyl, 2012*).

The strong effect of host species identity on the placement of individuals within isotopic space may be driven by a high degree of host-specificity in these interactions. For instance, a recent study by *Easson & Thacker (2014)* showed that microbial community diversity

within 20 HMA and LMA species (including 9 species from the current study) was strongly influenced by host phylogeny. More importantly, even closely related host species had strikingly different microbial communities, implying a strong selection for divergent microbial communities across sympatric sponge species and suggesting a potential role of microbial symbionts in niche differentiation (*Easson & Thacker, 2014*). Although we did not evaluate the microbial community composition in sponges from the Miskito Cays, additional work directly investigating how microbial community composition (*Webster et al., 2010*; *Webster & Taylor, 2012*; *Schmitt et al., 2012*) and function (*Stat, Morris & Gates, 2008*; *Thomas et al., 2010*; *Radax et al., 2012*) impact the placement of sponges in isotopic niche space is certainly warranted.

Coral reefs contain a staggering diversity of potential sources of C and N that are available to sponges, and our understanding of these sources, how they change across sites, and their respective isotope values is still being expanded (*Freeman & Thacker, 2011*; *Maldonado, Ribes & van Duyl, 2012*; *de Goeij et al., 2013*). With so many sources, some of which may have overlapping isotope values, it is difficult to construct effective and accurate mixing models, especially with only two isotopes (*Layman et al., 2012*). Thus, instead of estimating the relative contribution of nutrient sources to each holobiont, we used standardized methods to visualize the relative position of diverse holobionts within the isotopic niche space of oligotrophic reefs, providing an important baseline to which future research can be compared. For instance, across gradients from pristine to impacted or tropical to temperate reefs, we might hypothesize that the abundance and diversity of potential nutrient sources would increase, potentially relaxing holobiont dependence on symbiont metabolism and resulting in increased overlap in the $SEA_c$ of HMA and LMA groups, as well as of individual species in isotopic niche space (*Turner, Collyer & Krabbenhoft, 2010*; *Layman et al., 2012*). Research using these methods to understand how sponge communities and individual species respond to environmental gradients is ongoing as part of the Smithsonian Institution's Marine Global Earth Observatories (MarineGEO) initiative. Conducting similar studies in unison with community-based metrics outlined by *Layman et al. (2007a)* may also help us understand how dominant functional groups partition available resources within reef systems and the community-wide response to environmental change (*Layman et al., 2007b*).

In conclusion, we show that diverse sponge species fill available isotopic niche space within oligotrophic reefs and that, by hosting abundant symbiont communities, sponges are able to expand beyond the isotopic space occupied by LMA species. The placement of individual sponges within this isotopic space, however, may be driven more by the presence or absence of particular microbial taxa than by overall symbiont or photosymbiont abundance (*Thacker & Freeman, 2012*). We posit that, as these interactions have evolved, the acquisition and establishment of physiologically unique symbionts may have enabled some host species to expand into novel physiochemical niches (*Moran, 2007*; *Taylor et al., 2007*; *Thacker & Freeman, 2012*). Such expansion may support the coexistence of diverse sponge taxa within crowded reef systems.

## ACKNOWLEDGEMENTS

We thank the staff of the Smithsonian Marine Station for their logistical support. In particular, we thank S Reed and M Teplitski for their assistance in the field and S Box for technical support. We also thank V Paul and two reviewers for their helpful comments. M Zhu and H Wong assisted with isotope analyses. This is contribution number 971 from the Smithsonian Marine Station at Fort Pierce.

### Funding

Financial support for this project came from the Smithsonian Institution's Marine Global Earth Observatory (MarineGEO) and Tennenbaum Marine Observatories Network (TMON) initiatives. The funders had no role in study design, data collection and analysis, decision to publish, or preparation of the manuscript.

### Grant Disclosures

The following grant information was disclosed by the authors:
Smithsonian Institution's Marine Global Earth Observatory (MarineGEO).
Tennenbaum Marine Observatories Network (TMON) initiatives.

### Competing Interests

Christopher J. Freeman is an employee of the Smithsonian Marine Station.

### Author Contributions

- Christopher J. Freeman conceived and designed the experiments, performed the experiments, analyzed the data, contributed reagents/materials/analysis tools, wrote the paper, prepared figures and/or tables, reviewed drafts of the paper.
- Cole G. Easson performed the experiments, analyzed the data, contributed reagents/materials/analysis tools, wrote the paper, prepared figures and/or tables, reviewed drafts of the paper.
- David M. Baker analyzed the data, contributed reagents/materials/analysis tools, prepared figures and/or tables, reviewed drafts of the paper.

### Field Study Permissions

The following information was supplied relating to field study approvals (i.e., approving body and any reference numbers):

None of the sponge species collected from the Miskito Cays are under CITES protection. In addition, none of these species are protected in Honduras. These collections were carried out in collaboration with and under the oversight of the Centre for Marine Studies (CEM), a Honduran non-governmental organization that conducts applied research to underpin the management and sustainable use of marine biodiversity. The trip and collections were organized and authorized through an agreement between CEM in Honduras and the national government, with the Smithsonian providing technical support.

## Supplemental Information

Supplemental information for this article can be found online at http://dx.doi.org/10.7717/peerj.695#supplemental-information.

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
