# Peer review of "Metabolic diversity and niche structure in sponges from the Miskito Cays, Honduras"

_PeerJ, doi:10.7717/peerj.695_

## Round 0.1 · original submission · Minor Revisions

Both reviewers enjoyed you MS, but felt it needed modest revision before acceptance. I’ll send an email with a pdf containing some of my comments as well. Both reviewers provide specific comments that you should address in your revision. In general, both also thought the MS was too long for the story presented; I agree. Please revise to prevent redundancies, to get Introduction, M&M, Results, and Discussion portions each in their best location, and to focus on your main findings without too many side topics that may distract a reader. Both reviewers provides a lost of specific comments or suggestions. I summarize some of these below. When you revise the MS, please send a note outlining how you addressed each reviewer comment.

Reviewer #1:
Reviewer #1 notes, “Some information presented in the ‘M&M’ section seems like it belongs in the Introduction and similarly, discussions and interpretations in the Results section should probably be moved to the Discussion. The Discussion in particular seems too long given the data presented.” I agree. Please revise to focus on the major topics, to prevent redundancies, and to shorten the text. The more focused, direct, and tight the text is, the more readers will use it.

Please clearly explain sample sizes. Was there only one water sample taken at each location?

The reviewer asks for a list of species. See my comment on your text near Fig. 1. This would be a good place to add species names.

The Reviewer’s point #4 – I think your listing of latitude and longitude will allow more precise location of sites than will a general map, so you need not follow this request.


Reviewer #2:
This reviewer also felt the text could be tightened and shortened – please do so.

Reviewer 1 ·

Basic reporting

This is a descriptive paper, which describes the isotopic niche space and qualitative microbial abundance associated with 25 sponge species at 14 sites within the Miskito Cays, a relatively un-impacted coral reef system. The authors also compare isotopic values of POM in water samples collected from the same sites. The main findings are that chlorophyll concentrations vary substantially among sponge species sampled, species occupy different bivariate isotopic space depending upon both the qualitative abundances of associated microbes and, with respect to those species allocated to the group with high microbial abundances (HMA), this was also influenced by whether they had high or low levels of chlorophyll. They also found that there was a high degree of species-specificity within those groups.
The paper is interesting, but seems a little excessive in length given the data presented. Some information presented in the ‘M&M’ section seems like it belongs in the Introduction and similarly, discussions and interpretations in the Results section should probably be moved to the Discussion. The Discussion in particular seems too long given the data presented.

Experimental design

It is not clear why only one water sample was collected per site, whereas three sponge samples were collected per species per site (where available). More information on the sampling protocol and design would be advantageous.

I am not expert in the analyses used here so cannot comment on their appropriateness nor correct use, however, the bivariate plots are easy to interpret without analyses and the findings and conclusions seem valid.

Validity of the findings

I am not expert in the analyses used here so cannot comment on their appropriateness nor correct use, however, the bivariate plots are easy to interpret without analyses and the findings and conclusions seem valid. there seems to be a clear and consistent separation between HMA and LMA species, which for HMA species, is further complicated by chlorophyll levels.

Additional comments

Review of ”Metabolic diversity and niche structure in sponges from the Miskito Cays, Honduras”, by Freeman et al.
This is a descriptive paper, which describes the isotopic niche space and qualitative microbial abundance associated with 25 sponge species at 14 sites within the Miskito Cays, a relatively un-impacted coral reef system. The authors also compare isotopic values of POM in water samples collected from the same sites. The main findings are that chlorophyll concentrations vary substantially among sponge species sampled, species occupy different bivariate isotopic space depending upon both the qualitative abundances of associated microbes and, with respect to those species allocated to the group with high microbial abundances (HMA), this was also influenced by whether they had high or low levels of chlorophyll. They also found that there was a high degree of species-specificity within those groups.
The paper is interesting, but seems a little excessive in length given the data presented. Some information presented in the ‘M&M’ section seems like it belongs in the Introduction and similarly, discussions and interpretations in the Results section should probably be moved to the Discussion. The Discussion in particular seems too long given the data presented. I am not familiar with the analyses used here so cannot comment on their appropriateness nor correct use, however, the bivariate plots are easy to interpret without analyses and there seems to be a clear and consistent separation between HMA and LMA species, which for HMA species, is further complicated by chlorophyll levels.
Specific comments:
1. Minor comment, but L 100 – 105 This may be better suited to the Introduction to help outline the context for the study.
2. How many water samples were collected per site? Only one? But three sponge samples were collected per species per site (where available)? More information on the sampling protocol would be advantageous.
3. I think a list of the species in the main manuscript would be beneficial, given that abbreviations are used throughout. Such a list could be included within another Figure to save space/figure numbers.
4. A table with latitudes and and longitudes is perhaps less informative for the reader (particularly one with no knowledge of the study area) than a map showing locations.
5. Minor comment but some within L 232 – 236 belongs in Discussion.

Reviewer 2 ·

Basic reporting

In this manuscript, Freeman et al apply new analytical tools to study the isotopic niche space of sponges. The manuscript is well written and the discussion and conclusions are generally well placed according to the results.

Experimental design

The study is performed based on 14 different collection sites and 19 sponge species, making it comprehensive. The methods are sound and well described.

Validity of the findings

The comparison of isotopic niche space between sponges is performed taking in consideration the HMA/LMA dichotomy and whether sponges carry or not photosymbionts, which add valuable biological aspects to the discussion of the results.

Line 90: The abundance of photosymbionts seems to be an important factor related to the placement of species in isotopic space, specially considering figure 3c. Is it possible to test which grouping factor (i.e. HMA/LMA or presence/absence of photosymbionts) correlates better with the placement of samples in the isotopic space? If possible, such an approach will add an interesting perspective to the current manuscript.

Lines 371-373: The current manuscript verifies the influence of the microbial abundance and photosymbiont abundance in the placement of species in isotopic space, but do not investigate the influence of the taxon composition of sponge symbionts. The hypothesis that the former may be more influential in the placement of species in isotopic space than the first two factors is not direct and requires more argumentation.

Additional comments

The introduction is lengthy and the authors should consider reducing it and eliminating redundancy. As suggestion, paragraphs of the lines 25-36 and 61-82 could be merged. Methods described in lines 76-82 are redundant with lines 138-139/154-156. If authors want to mention the recent analytical advances in the field (lines 76-82), the description of methods should be accompanied by their impact on the understanding of ecological processes.

My minor comments are:

L5: The notion that LMA sponges do not host symbionts is not correct. The authors should review this sentence in light of the recent findings of Giles et al FEMS Microbiology Ecology 2013 and Moitinho-Silva et al Molecular Ecology 2014.
L60: The statement that sponges host "almost all evolutionary lineages of bacteria and archaea" may be exaggerated. Authors should review this sentence.
L153: Please provide further details of how the normality and equal variance were tested.
L246: The word "unfortunately" may be adding an unnecessary sense of expectation in this case.
L284-286: What about other microbial metabolic pathways that were described in sponges and are not mentioned here, e.g. the chemoautotrophic processes listed by Taylor et al Microbiology and Molecular Biology Reviews 2007?
L335-337: I think this is an important interpretation the results. It may be mentioned earlier in the discussion to call more attention. Is there an experimental set-up to distinguish the placement of species in isotopic space due to the utilization of nutrient sources from the placement of species due to cycling of nutrients within the holobiont?
L369: This sentence leads the reader to the notion that LMA are devoid of symbionts. It should be reformulated. Please consider the comment of the line 5.
L382: Please inform the contribution number

---

## Round 0.2 · accepted · Accept

You did a fine job of dealing with reviewer comments. I'm happy to accept the MS.